# Epidemiological analysis of intramuscular hemorrhage of respiratory and accessory respiratory muscles in fatal drowning cases

**Daiko Onitsuka[1,2], Takuma Nakamae[1], Midori Katsuyama[1], Machiko Miyamoto[1], Eri Higo[1], Masahiko Yatsushiro[1], Takahito Hayashi[1] ***

1 Department of Legal Medicine, Graduate School of Medical and Dental Sciences, Kagoshima University, Kagoshima, Japan, 2 Kushikino Coast Guard Office, Tenth Regional Coast Guard Headquarters, Japan Coast Guard, Kagoshima, Japan

* takahito@m2.kufm.kagoshima-u.ac.jp

**Data Availability Statement:** All relevant data are within the manuscript and its Supporting Information files.

## Abstract

The postmortem diagnosis of drowning death and understanding the mechanisms leading to drowning require a comprehensive judgment based on numerous morphological findings in order to determine the pathogenesis and epidemiological characteristics of the findings. Effortful breathing during the drowning process can result in intramuscular hemorrhage in respiratory and accessory respiratory muscles. However, the characteristics of this phenomenon have not been investigated. We analyzed the epidemiological characteristics of 145 cases diagnosed as drowning, in which hemorrhage, not due to trauma, was found in the respiratory muscles and accessory respiratory muscles. Hemorrhage was observed in 31.7% of these cases, and the incidence did not differ by gender or drowning location. The frequency of hemorrhage was significantly higher in months with a mean temperature below 20˚C than in months above 20˚C, suggesting a relationship between the occurrence of hemorrhage and low environmental temperature. Moreover, the frequency of hemorrhage was significantly higher in the elderly (aged ≥65 years) compared to those <65 years old. In the elderly, the weakening of muscles due to aging may contribute to the susceptibility for intramuscular hemorrhage. Moreover, these intramuscular hemorrhages do not need to be considered in cases of a potential bleeding tendency due to disease such as cirrhosis or medication such as anticoagulants. Our results indicate that intramuscular hemorrhage in respiratory and accessory respiratory muscles can serve as an additional criterion to differentiate between fatal drowning and other causes of death, as long as no cutaneous or subcutaneous hematomas above the muscles with hemorrhages are observed. In addition, the epidemiological features that such intramuscular hemorrhage is more common in cold environments and in the elderly may provide useful information for the differentiation.

## Introduction

In forensic autopsy practice, diagnosing the cause of death of immersed corpses is a very difficult task. In the case of immersed corpses, not all deaths are due to drowning, and many

**Funding:** this work was supported by JSPS KAKENHI Grant Number 18K10133. The full name of the funder is Japan Society for the Promotion of Science (https://www.jsps.go.jp/index.html). The funders had no role in study design, data collection and analysis, decision to publish, or preparation of the manuscript.

**Competing interests:** All authors declare no conflict of interest related to this study.

circumstances are possible, from natural death to homicide. In addition, even if the cause of death is drowning, the cause of drowning can range from external, such as trauma or the effects of drugs, to internal causes [1, 2]. Therefore, in the case of corpse in water, the diagnosis of the cause of death should be made comprehensively and carefully on the basis of autopsy and supplemental examination findings [1–4].

The most important external sign of vital drowning is froth, which results from the mixture of air, water of the drowning medium, proteinaceous exudate, and pulmonary surfactant [5, 6]. The froth is observed typically as a fungiform structure (mushroom shape) around the mouth and nostrils, but is only seen in fresh corpses (within about 1 day of death) [4]. More-over, it is not specific to drowning, as froth is also present in other causes of death, such as cardiogenic pulmonary edema, poisoning, and strangulation [3, 4]. The most important internal sign of drowning is hyperinflation of the lungs (called emphysema aquosum) [1–6]. Increased phlegm secretion caused by aspiration of drowning medium and froth formation create a valve mechanism that traps air in the airways, resulting in the ballooned lungs filling the thoracic cavity when the sternum is removed at autopsy. In such hyperinflated lungs, imprints of the ribs on the lateral surface of the lungs can often be observed [1, 5, 6]. Another finding of pulmonary hyperinflation are so-called Paltauf's spots, boundary indistinct and pale hemor-rhages per rhexis under the pleura [7]. Findings suggestive of vital drowning in other organs include severe congestion and hemorrhages within the petrous temporal bones [1], longitudi-nal laceration in gastric mucosa [4, 8, 9], and separation of the gastric contents into three layers (called Wydler's sign, contents are separated into foamy, drowning medium, and food particle phases) [4, 10, 11].

Despite the numerous morphological findings suggesting drowning, the postmortem diag-nosis of death by drowning based on autopsy findings is difficult and remains a diagnosis of exclusion [12]. None of these findings alone are specific enough to diagnose death by drown-ing [13]. Furthermore, in corpses with advanced putrefaction, the above findings are absent, making it difficult to diagnose vital drowning [14]. Therefore, the postmortem diagnosis of drowning death and the mechanism leading to drowning requires a comprehensive judgment based on the abovementioned morphological findings. Therefore, it is very important to inves-tigate the pathogenesis and epidemiological characteristics of the findings.

Diatoms (planktons) test is also available as supplementary examinations for postmortem diagnosis of drowning death [15]. In vital drowning case, diatoms dwelling in the drowning medium can be detected in the systemic circulatory organs including the spleen, liver, kidneys, and bone marrow. However, the detection of diatoms has traditionally been critically ques-tioned, as diatoms are ubiquitous and have been detected in the organs, especially in the bone marrow (femur), of corpses with a cause of death other than drowning [16, 17].

Hemorrhage in respiratory and accessory respiratory muscles, not originating from trauma, is observed commonly in fatal drowning cases [1–4, 7, 18, 19]. Although different mechanisms of intramuscular hemorrhage have been described between studies, the rupture of fascia due to repeated contraction and relaxation of muscles during effortful breathing that occurs during the process of drowning leads to subfascial and intramuscular hemorrhage. However, the epi-demiological characteristics of intramuscular hemorrhage have not been investigated. In this study, we investigated the relationship between the frequency of intramuscular hemorrhage and the age, gender, drowning location, and environmental temperature of the drowning case.

## Materials and methods

Among the forensic autopsy cases performed at the Department of Legal Medicine, Graduate School of Medical and Dental Sciences, Kagoshima University between January 1, 2009 and

May 31, 2020, we examined 145 cases (87 males and 58 females; mean age, 60.8 years; median age, 64.0 years; range in age, 4 to 89 years; postmortem interval, 18 to 72 hours) diagnosed as drowning, excluding cases with advanced postmortem decomposition. We retrospectively examined the gender, age, location of drowning, season, average temperature in the month of discovery, and effect of a potential bleeding tendency due to disease (e.g., liver cirrhosis) or medication (e.g., anticoagulants, antiplatelet agents) in the cases of hemorrhage in the respiratory and accessory respiratory muscles of the neck and chest that were not due to injury or cardiac massage therapy based on autopsy records and information from police investigations. The relationship between the number of muscles with hemorrhage and these epidemiological factors was also examined. The number of muscles was counted as 1 if hemorrhage was observed unilaterally. Although there are accessory respiratory muscles in the back such as trapezius, supraspinatus, and infraspinatus, they were excluded from the present study because the muscles of back were not examined in all cases.

Statistical analysis was performed using the Mann-Whitney $U$ test for comparisons between two groups and one-way analysis of variance (Tukey-Kramer method as post-hoc test) for comparisons between multiple groups. The correlation coefficient of temperature and age was determined using Spearman's correlation coefficient by rank test. A value of $p<0.05$ was considered to indicate a significant difference.

This study was approved by the Ethics Committee for Epidemiological Research, Graduate School of Medical and Dental Sciences, Kagoshima University (Approval No. 200247) and was carried out in accordance with the Declaration of Helsinki Principles. Moreover, this study was conducted using autopsy records from the past, and we could not obtain informed consent from the bereaved family for the use of the records. Therefore, in accordance with the "Ethical Guidelines for Medical Research Involving Human Subjects (enacted by the Ministry of Health, Labor and Welfare in Japan)," Section 12–1 (2) (a) (c), information on the implementation of the study was posted on our website (http://www.kufm.kagoshima-u.ac.jp/~legalmed/), and if there was a request to refuse the use of the samples for research, they were excluded from samples of this study. In addition, the Ethics Committee for Epidemiological Research, Graduate School of Medical and Dental Sciences, Kagoshima University (Approval No. 200247) has approved the waiver for the informed consent of this study.

## Results

In our retrospective review of 145 forensic autopsy cases classified as drowning, 46 cases (31.7%) showed bleeding in respiratory or accessory respiratory muscles (Fig 1). Muscles with hemorrhage include the intercostal muscles, greater pectoral muscles, smaller pectoral muscles, platysma muscles, sternocleidomastoid muscles, sternohyoid muscles, sternothyroid muscles, and omohyoid muscles. Of the neck accessory respiratory muscles, the sternocleidomastoid muscles showed evidence of hemorrhage most frequently (n = 19, 13%), while the greater pectoral muscles showed hemorrhage most frequently compared to other thoracic muscles (n = 29, 20%). There was no significant difference observed in the extent of hemorrhage between bilateral muscles (Table 1).

### Gender and age

In regards to gender, hemorrhage in any muscles was observed in 26 of 87 male (30%) and 20 of 58 female (34%) cases, with no significant difference ($p$ = 0.5614, Fig 2A). There was also no significant difference by gender in the number of muscles with hemorrhage ($p$ = 0.6138, Fig 2B).

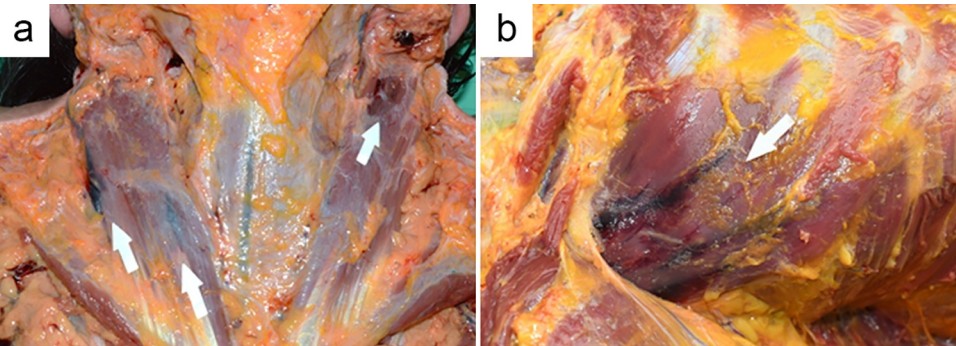

**Fig 1. Representative photographs of intramuscular hemorrhage in accessory respiratory muscles obtained from study samples.** (a) Sternocleidomastoid muscle of female in her late 60s. (b) Pectoralis minor muscle of male in his early 70s.

We divided the drowning cases into age groups and examined differences in hemorrhage among the groups. There were 2 of 11 cases with intramuscular hemorrhage in the group aged 0–20 years (frequency, 0.18), 3 of 12 cases in the group aged 21–30 years (frequency, 0.25), 4 of 11 cases in the group aged 31–40 years (frequency, 0.36), 3 of 16 cases in the group aged 41–50 years (frequency, 0.19), 11 of 44 cases in the group aged 51–60 years (frequency, 0.25), 13 of 28 cases in the group aged 61–70 years (frequency, 0.46), and 10 of 23 cases in the group aged 71–80 years (frequency, 0.43). No significant difference was observed in hemorrhage among the age groups ($p = 0.2645$), however, there was a tendency for the elderly cases to have a higher frequency of hemorrhage. We divided the cases into two groups: those <65 years old (n = 76) and those ≥65 years old (n = 69). As a result, the frequency of intramuscular hemorrhage was significantly higher ($p = 0.0295$) in cases aged ≥65 years than in those aged <65 years (Fig 3A, frequency, 0.41 vs. 0.24). The number of muscles with hemorrhage was also significantly higher ($p = 0.00945$) in victims ≥65 years than in those aged <65 years (Fig 3B, 1.0 vs. 0.38).

## Drowning location

Drowning sites were classified into four groups: sea (96 cases, 58 ± 2 years old), river (33 cases, 66 ± 3 years old), ditches/irrigation channel (4 cases, 77 ± 4 years old), and bathtub (12 cases, 67 ± 7 years old). Cases found near estuaries (in brackish water area) were excluded from the samples. The frequency of intramuscular hemorrhage was similar in cases drowning in the sea (frequency, 0.31) and in rivers (frequency, 0.33), slightly higher in cases drowning in ditches/irrigation channel (frequency, 0.50), and lower in cases drowning in bathtubs (frequency, 0.25), but there was no significant difference among the four groups ($p = 0.826$, Fig 4A).

**Table 1. Muscles with intramuscular hemorrhage.**

| Muscle | Cases (frequency) | Left (frequency) | Right (frequency) | *p*-value |
|---|---|---|---|---|
| Intercostal muscle | 3 (0.02) | 2 (0.01) | 3 (0.02) | 0.652 |
| Platysma muscle | 2 (0.01) | 1 (0.006) | 1 (0.006) | 1.000 |
| Sternocleidomastoid muscle | 19 (0.13) | 14 (0.097) | 13 (0.09) | 0.919 |
| Sternohyoid muscle | 14 (0.10) | 11 (0.076) | 10 (0.069) | 0.821 |
| Sternothyroid muscle | 11 (0.08) | 4 (0.028) | 10 (0.069) | 0.100 |
| Omohyoid muscle | 2 (0.01) | 0 (0) | 2 (0.014) | 0.157 |
| Greater pectoral muscle | 18 (0.12) | 12 (0.08) | 14 (0.1) | 0.682 |
| Smaller pectoral muscle | 29 (0.20) | 22 (0.15) | 17 (0.12) | 0.390 |

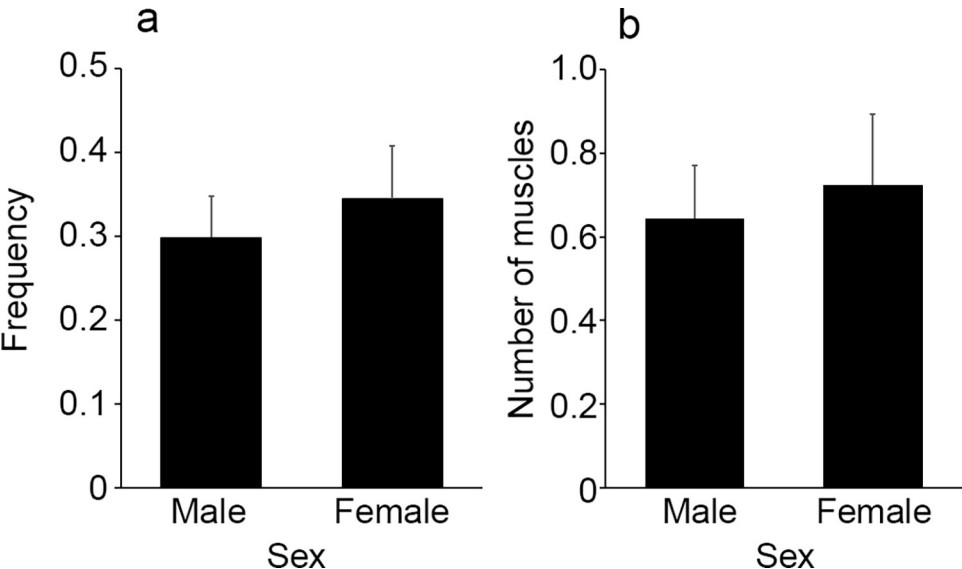

**Fig 2.** Comparison of the frequency of intramuscular hemorrhage (a) and the number of muscles with hemorrhage (b) by gender. All values represent the mean ± SEM. No statistically significant differences due to gender were noted.

Similar results were observed in the number of muscles with hemorrhage observed among the four groups ($p = 0.964$, Fig 4B).

## Seasonal and monthly mean temperature

The frequency of intramuscular hemorrhage observed in drowning cases was highest in March (13 of 17 cases, frequency, 0.76), followed by December (4 of 7 cases, frequency, 0.57) and

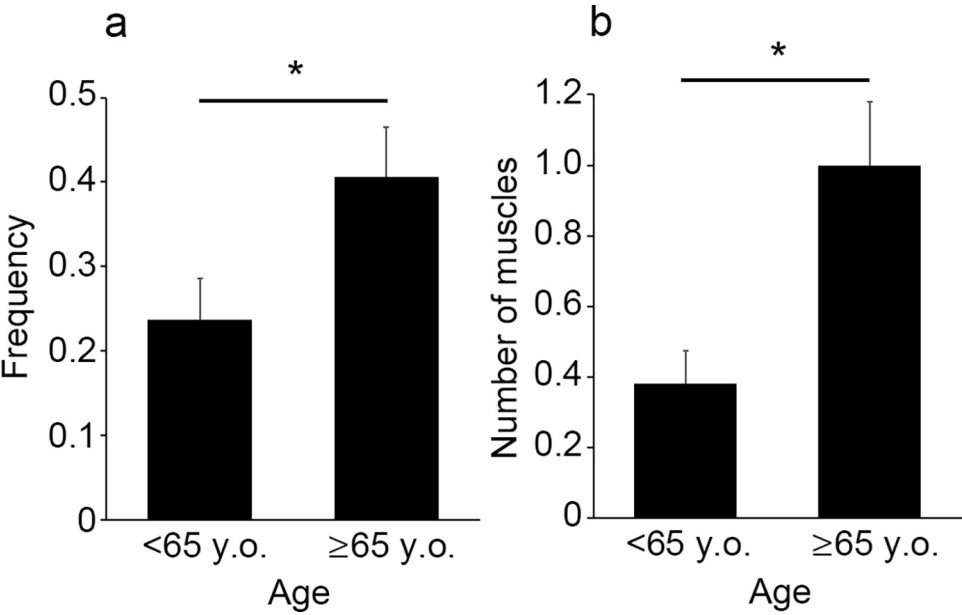

**Fig 3.** Comparison of the frequency of intramuscular hemorrhage (a) and the number of muscles with hemorrhage (b) between victims <65 years old and ≥65 years old. All values represent the mean ± SEM. *$p < 0.05$.

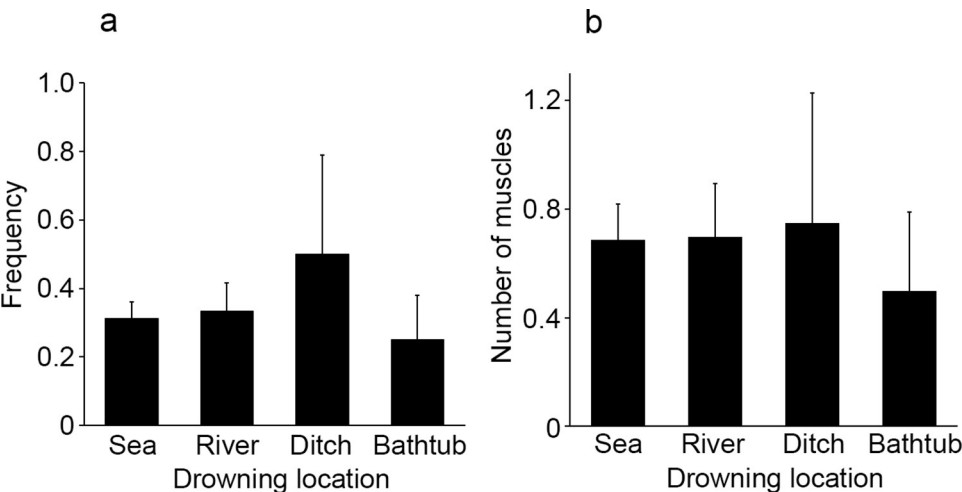

**Fig 4.** Comparison of the frequency of intramuscular hemorrhage (a) and the number of muscles with hemorrhage (b) at the drowning site. All values represent the mean ± SEM.

November (6 of 17 cases, frequency, 0.35) (Table 2). As for the number of muscles with hemorrhage, March was also the month with the most muscles showing hemorrhage (mean, 1.82), followed by December (mean, 1.71) and November (mean, 0.82) (Table 2).

The frequency of intramuscular hemorrhage was 0.47 in the spring (March-May; frequency, 0.47), 0.16 in the summer (June-August; frequency, 0.16), 0.26 in the fall (September-November; frequency, 0.26), and 0.36 in the winter (December-February; frequency, 0.36). A significant difference was observed only between the spring and summer seasons ($p = 0.0227$, Fig 5A). The number of muscles with hemorrhage was 0.98 in the spring, 0.27 in the summer, 0.58 in the fall, and 0.88 in the winter. A significant difference was observed in the number of muscles with hemorrhage only between drowning cases between spring and summer ($p = 0.050$, Fig 5B). When we examined the results by season, we found that the frequency of hemorrhage in drowning cases was higher in the winter and spring, with the number of muscles with hemorrhage tending to be larger in those seasons.

According to the monthly mean temperature (statistical period, 2009–2020) of Kagoshima City, which is located in the center of Kagoshima Prefecture (excluding outlying islands) [20],

**Table 2. Monthly frequency of intramuscular hemorrhage.**

| Month (cases) | Cases with hemorrhage (frequency) | Number of muscles |
|---|---|---|
| January (9) | 3 (0.33) | 0.56 |
| February (9) | 2 (0.22) | 0.56 |
| March (17) | 13 (0.76) | 1.82 |
| April (16) | 2 (0.13) | 0.38 |
| May (12) | 6 (0.5) | 0.58 |
| June (7) | 1 (0.14) | 0.43 |
| July (16) | 2 (0.13) | 0.13 |
| August (14) | 3 (0.21) | 0.36 |
| September (10) | 2 (0.20) | 0.30 |
| October (11) | 2 (0.18) | 0.45 |
| November (17) | 6 (0.36) | 0.82 |
| December (7) | 4 (0.57) | 1.71 |

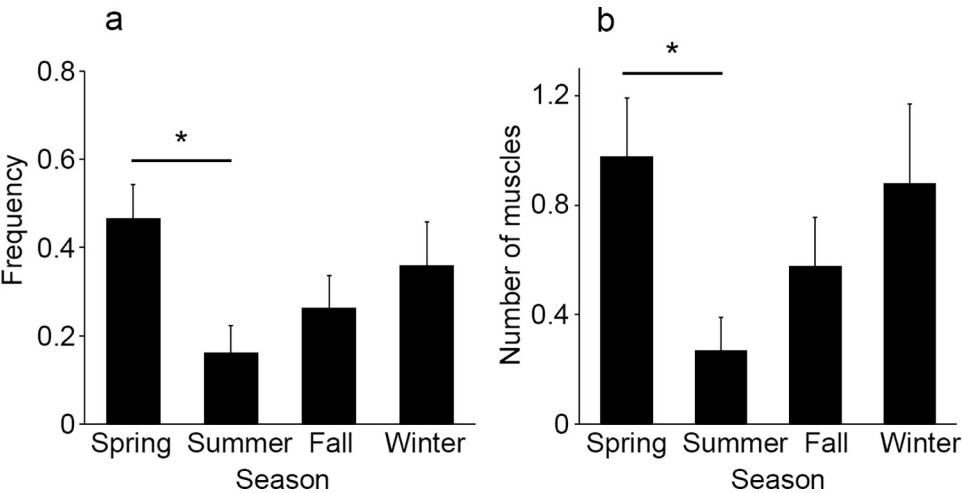

**Fig 5.** Comparison of the frequency of intramuscular hemorrhage (a) and the number of muscles with hemorrhage (b) by season. All values represent the mean ± SEM. *$p < 0.05$.

the frequency of intramuscular hemorrhage was significantly higher in months when the temperature was below 20°C (75 patients, 65 ± 2 years old) than in months when the temperature was above 20°C (70 patients, 57 ± 3 years old) ($p = 0.0271$, 0.4 vs. 0.23, Fig 6A). The number of bleeding muscles was also significantly higher in months with temperatures below 20°C than in months above 20°C ($p = 0.0093$, 0.97 vs. 0.36, Fig 6B). There was no significant correlation between temperature and age ($p = 0.116183$).

The temperature of seawater, river water, and water in ditches/irrigation canals correlated with the seasonal temperature, but the temperature of water in the bathtub did not correlate with seasonal temperature. Accordingly, we conducted the same analysis as above in the 133 cases whose place of discovery was not the bathtub. We found that the frequency of

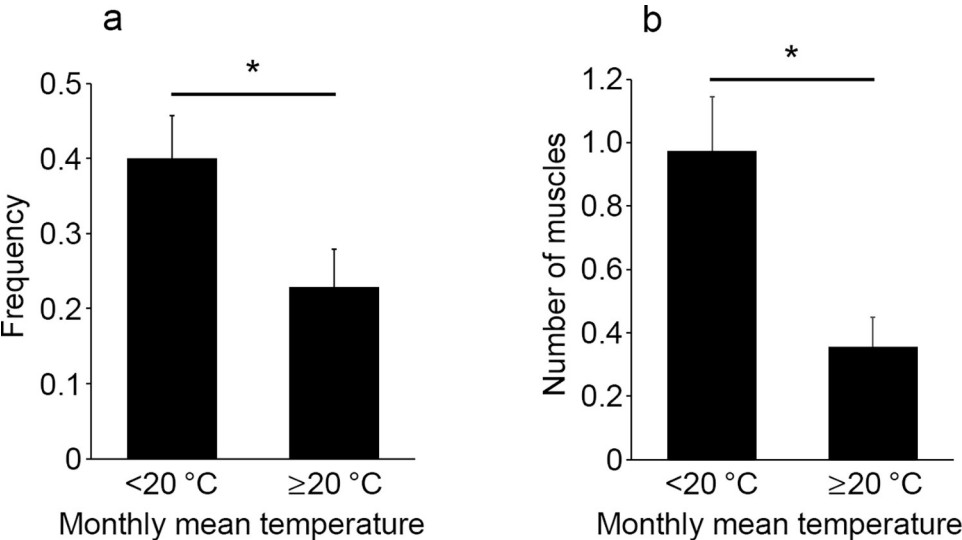

**Fig 6.** Comparison of the frequency of intramuscular hemorrhage (a) and the number of muscles with hemorrhage (b) between monthly mean temperature below 20°C and monthly mean temperature above 20°C. All values represent the mean ± SEM. *$p < 0.05$.

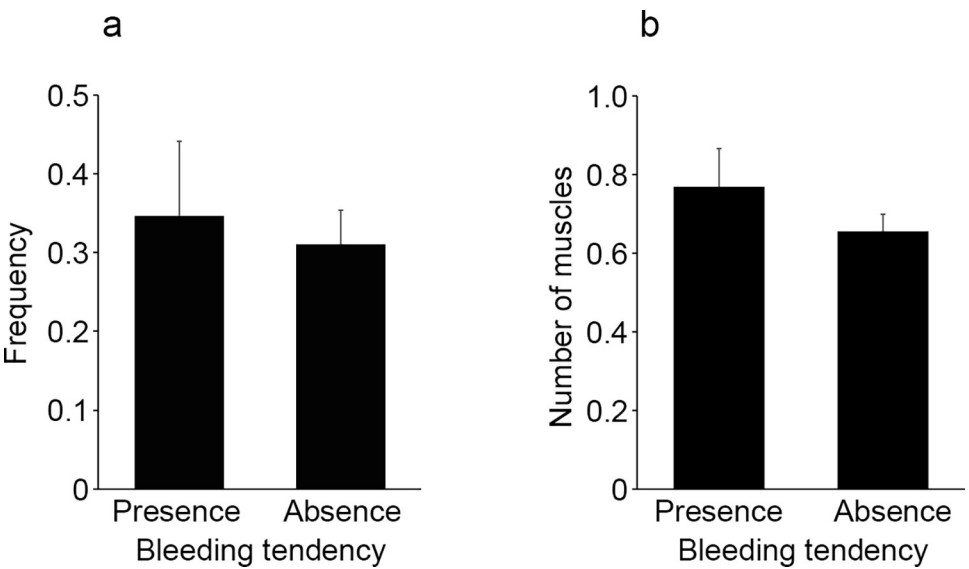

**Fig 7.** Comparison of the frequency of intramuscular hemorrhage (a) and the number of muscles with hemorrhage (b) between victims with or without a potential bleeding tendency. All values represent the mean ± SEM. $^{*}p{<}0.05$.

intramuscular hemorrhage was significantly higher ($p$ = 0.0486, 0.40 vs. 0.24, S1A Fig) and the number of muscles with hemorrhage was significantly higher ($p$ = 0.01701, 1.0 vs. 0.38, S1B Fig) in months when the mean monthly temperature was less than 20˚C (67 cases, 57 ± 2 years old) compared to months when the mean monthly temperature was 20˚C or higher (66 cases, 64 ± 2 years old).

## Effects of a potential bleeding tendency

Twenty-six cases (17.9%) with a potential bleeding tendency due to disease (liver cirrhosis, n = 8) or medication (anticoagulants or antiplatelet agents, n = 19) were included in this study. There was no significant difference in the frequency of intramuscular hemorrhage according to the presence or absence of a potential bleeding tendency ($p$ = 0.726589, Fig 7A). Similar results were obtained for the number of muscles with hemorrhages ($p$ = 777439, Fig 7B).

## Discussion

Intramuscular hemorrhage of the sternocleidomastoid and greater pectoral muscles in fatal drowning cases was first reported by Paltauf, et al. [7]. Later, Reuter reported that 11.5% [18] or 13.2% [19] of drowning cases had hemorrhage in the respiratory and accessory respiratory muscles, and the author argued that such intramuscular hemorrhage was due to spasmodic dyspnea (effortful breathing) occurring during the drowning process. Püschel et al. [21] found that 51.3% (20 out of 39) of drowning cases showed hemorrhage in respiratory and accessory respiratory muscles, as well as other back and upper extremity muscles, suggesting that such hemorrhage may be caused not only by spasmodic dyspnea, but also by systemic convulsions during the agonal stage in the process of asphyxiation or by "struggling" to avoid drowning. Recently, Oshima et al. [22] reported that 63.4% (104 out of 164) of drowning cases showed hemorrhage in the muscles around the scapula (supraspinatus, infraspinatus, and rhomboid muscles), and that agonal convulsions may not cause such hemorrhage, but by intentional violent movement of the upper limbs in the act of struggling to avoid drowning (self-rescue attempts). In the present study, intramuscular hemorrhage was observed in 31.7% (46 out of

145) of the drowning cases, but it was limited to the respiratory and accessory respiratory muscles of the neck and chest. While the number of cases and types of muscles vary between studies, and opinions differ on the mechanism of intramuscular hemorrhage, the finding of intramuscular hemorrhage in 11.5% to 63.4% of drowning cases reaffirms that hemorrhage is a useful auxiliary finding for the forensic diagnosis of drowning. However, it should be noted that such intramuscular hemorrhage in cases of fatal hanging and strangulation has already been described and was assumed to be an asphyxia-associated finding [23, 24]. Moreover, we have included in our study those cases in which there was no hematomas of skin and subcutaneous fatty tissue above the muscle in which the hemorrhage was found. In actual forensic autopsy practice, the possibility of blunt trauma as a cause of the observed intramuscular hemorrhage must be ruled out.

Püschel et al. [21] reported no significant correlation between age and the frequency of intramuscular hemorrhage. However, in the present study, the frequency of intramuscular hemorrhage and the number of muscles with hemorrhage were significantly higher in patients ≥65 years of age compared to those <65 years of age (Fig 3), which suggests that intramuscular hemorrhage may be more likely to occur in the elderly cases of drowning. Aging causes an imbalance between the synthesis and degradation of muscle constituent proteins, leading to atrophy and loss of myofibers, which in turn makes the muscle itself vulnerable to damage [25, 26]. Therefore, as described in detail in the previous paragraph, the repeated contraction and relaxation of muscles due to spasmodic dyspnea resulting from the drowning process, agonal convulsions, or "struggling" to avoid drowning may cause injury to muscles in elderly drowning cases. In the study by Oshima et al. [22], the reported frequency of intramuscular hemorrhage was significantly higher (63.4%) than that in any other study, possibly due to the large proportion of elderly subjects (median age, 73 years; range in age, 3 to 97 years). It should be noted that the mean age of the subjects in the present study was <65 years (mean age, 60.8 years; median age, 64.0 years; range in age, 4 to 89 years).

The frequency of intramuscular hemorrhage and the number of muscles with hemorrhage were not significantly different among the four drowning location groups: sea, river, ditch/ irrigation channels, and bathtub. In general, the frequency of intramuscular hemorrhage and the number of muscles with hemorrhage in freshwater drowning cases were expected to be lower because of the shorter time of death due to hyperkalemia caused by hemodilution and hemolysis [2]. However, there was no significant difference in the frequency of intramuscular hemorrhage or the number of muscles with hemorrhage in freshwater drownings compared to seawater drownings. On the other hand, in the study by Oshima et al. [22, 27], although no statistical analysis was performed, the frequency of intramuscular hemorrhage was similar in the sea, rivers, and ditch/ irrigation channels, with the highest frequency in ponds and the lowest in bathtubs. Bath-related drowning in Japan may be caused by loss of consciousness due to prior endogenous diseases such as arrhythmia caused by temperature change, cardiovascular disease, and the development of cerebrovascular disease [28–30]. The process of drowning in the bath is considered to be shorter than that in other places such as the sea and rivers. Therefore, the frequency of intramuscular hemorrhage, which is thought to occur in the process of drowning as described above, is also expected to be low. Schneppe et al. [4] also described that pre-existing conditions can shorten the time to death onset in cases of drowning of elderly individuals. However, the length of survival time (acute or prolonged death) cannot be estimated solely by autopsy findings. Accordingly, the relationship between the frequency of intramuscular hemorrhage and survival time could not be investigated in detail in this study.

To the best of our knowledge, the relationship between the frequency of intramuscular hemorrhage in fatal drowning cases and the season in which the drowning occurred has not been investigated. In our study, the frequency of hemorrhage was significantly higher and the

number of hemorrhaged muscles was significantly higher in months when the mean temperature was <20˚C than in months when the temperature was ≥20˚C. In general, except for bath-related drowning, the temperature of the water in drowning cases correlated with a higher frequency of hemorrhage and a higher number of hemorrhaged muscles. Thus, our results may lead to the conclusion that intramuscular hemorrhage is more likely to occur in drownings that occur under low temperature conditions. As there was no significant correlation between temperature and age, as low temperature was an independent factor in the likelihood of intramuscular hemorrhage. However, intramuscular hemorrhage is more likely to occur in low-temperature drowning because, in addition to the effortful breathing described above, drowning in a low-temperature environment causes body shivering to produce heat as a defensive reaction, i.e., repetition of violent muscle contraction and relaxation [31], which may cause intramuscular hemorrhage. We examined 57 cases in which the water temperature at the time of discovery was known (mean temperature, 21.2˚C; range, 6 to 46˚C) and divided the cases into two groups (<20˚C and ≥20˚C) based on the temperature of 20˚C, as well as the monthly mean temperature (frequency, $p = 0.0805$; muscle number, $p = 0.0909$; S2 Fig). Since it is difficult to accurately estimate the drowning location, drowning time, or time of death, and, therefore, difficult to identify the water temperature at the time of drowning accurately, we could not investigate the relationship between the water temperature at the time of drowning and the frequency of intramuscular hemorrhage.

Finally, our initial expectation was that intramuscular hemorrhage would be more frequent in cases with a potential bleeding tendency due to disease (liver cirrhosis) or medication (anticoagulants or antiplatelet agents). However, contrary to our expectation, the presence or absence of bleeding tendency had no significant effect on either the frequency of intramuscular hemorrhage or the number of muscles with hemorrhage. Therefore, such intramuscular hemorrhage of immersed corpses does not need to be considered as the influence of bleeding tendency.

## Conclusions

Our results indicate that the frequency of hemorrhage in the respiratory and accessory respiratory muscles in drowning cases was 31.7%, reaffirming intramuscular hemorrhage as an additional criterion to allow differentiation between fatal drowning and other causes of death, as long as no hematomas of skin and subcutaneous fatty tissue above the intramuscular hemorrhages can be observed and pre and post mortal traumatization can be ruled out completely. In low environmental temperatures, effortful breathing and body shivering caused by the low temperature may lead to intramuscular hemorrhage in fatal drowning. In addition, elderly drowning case ≥65 years old may be more prone to intramuscular hemorrhage in fatal drowning due to muscle weakening caused by aging. Furthermore, such intramuscular hemorrhage of immersed corpses does not need to be considered as the influence of bleeding tendency.

## Supporting information

**S1 Fig.** Comparison of intramuscular hemorrhage (a) and number of muscles with hemorrhage (b) between monthly mean temperature below 20˚C and above 20˚C, excluding cases of drowning in the bathtub. All values represent the mean ± SEM.
(TIF)

**S2 Fig.** Comparison of the frequency of intramuscular hemorrhage (a) and the number of muscles with hemorrhage (b) between water temperature (<20˚C and ≥20˚C) at the time of

discovery. All values represent the mean ± SEM.
(TIF)

## Author Contributions

**Conceptualization:** Eri Higo, Takahito Hayashi.

**Data curation:** Daiko Onitsuka, Takuma Nakamae, Midori Katsuyama.

**Funding acquisition:** Takahito Hayashi.

**Investigation:** Daiko Onitsuka, Takuma Nakamae, Midori Katsuyama, Machiko Miyamoto, Eri Higo, Masahiko Yatsushiro.

**Methodology:** Daiko Onitsuka, Takuma Nakamae, Midori Katsuyama, Machiko Miyamoto, Eri Higo, Masahiko Yatsushiro.

**Project administration:** Takahito Hayashi.

**Supervision:** Takahito Hayashi.

**Validation:** Midori Katsuyama, Masahiko Yatsushiro, Takahito Hayashi.

**Visualization:** Machiko Miyamoto, Eri Higo.

**Writing – original draft:** Daiko Onitsuka.

**Writing – review & editing:** Takahito Hayashi.

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
