## [Decision Letter · Decision Letter 0]

22 Sep 2021

PONE-D-21-27906Epidemiological analysis of intramuscular hemorrhage of respiratory and accessory respiratory muscles in fatal drowning casesPLOS ONE

Dear Dr. Hayashi,

Thank you for submitting your manuscript to PLOS ONE. After careful consideration, we feel that it has merit but does not fully meet PLOS ONE’s publication criteria as it currently stands. Therefore, we invite you to submit a revised version of the manuscript that addresses the points raised during the review process.

We look forward to receiving your revised manuscript.

Kind regards,

Stephen Raverty

Academic Editor

PLOS ONE

Journal Requirements:

2. In your ethics statement in the manuscript and in the online submission form, please ensure that you have discussed whether all data/samples/autopsy reports were fully anonymized before you accessed them and/or whether the IRB or ethics committee waived the requirement for informed consent. If next-of-kin provided informed written consent to have data from autopsies used in research, please include this information.

4. We note that Figure 1 in your submission contain copyrighted images. All PLOS content is published under the Creative Commons Attribution License (CC BY 4.0), which means that the manuscript, images, and Supporting Information files will be freely available online, and any third party is permitted to access, download, copy, distribute, and use these materials in any way, even commercially, with proper attribution. For more information, see our copyright guidelines: http://journals.plos.org/plosone/s/licenses-and-copyright.

Additional Editor Comments:

Both reviewers identified merit with your manuscript and provided valuable insights and suggestions to improve the content. Additional details have been requested for the introduction by one reviewer. Please review the comments and address accordingly in your revision. Thank you for submitting this manuscript to PLOS ONE

Reviewers' comments:

Reviewer's Responses to Questions

**Comments to the Author**

1. Is the manuscript technically sound, and do the data support the conclusions?

Reviewer #1: Yes

Reviewer #2: Partly

2. Has the statistical analysis been performed appropriately and rigorously? 

Reviewer #1: Yes

Reviewer #2: Yes

3. Have the authors made all data underlying the findings in their manuscript fully available?

Reviewer #1: Yes

Reviewer #2: Yes

4. Is the manuscript presented in an intelligible fashion and written in standard English?

Reviewer #1: Yes

Reviewer #2: Yes

5. Review Comments to the Author

Reviewer #1: Manuscript Number: PONE-D-21-27906

Title: Epidemiological analysis of intramuscular hemorrhage of respiratory and accessory respiratory muscles in fatal drowning cases

General comments

This study is the statistical analysis of intramuscular hemorrhage of respiratory and accessory respiratory muscles in fatal drowning cases. Based on previous research reports, this paper found that intramuscular bleeding had a significant difference in temperature and age. This is a very interesting report. However, some amendments are needed to make it public.

Major comments

1. Drowning in fresh water is known to cause premature death by fatal arrhythmia due to hemolysis of red blood cells and hyperkalemia.

The author should discuss the difference in intramuscular hemorrhage between freshwater and seawater drowning.　

2. In the cases of the river, it is a brackish water area at the mouth of the river, so it should separate from the river.

Minor comments

1. Please set the month corresponding to every seasons.

2. On page 6, line 126, “In patients” should be “in victims” or “in deceased”.

Reviewer #2: 1. The introduction is way to short because the diagnosis of drowning is a challenging issue with so much literature to introduce. The morphological findings, that leads to the diagnosis 'drowning' must be more explained. Moreover the difficulty of the diagnosis drowning are untouched. (see e.g. Schneppe et al 2021)

2. The detection of diatomes is questioned critically, that must be mentioned (e.g. Spitz, WU 1963, Schneider, V 1980) .

3. The authors conclude, that intramuscular hemorrhages are an auxilary finding for the diagnosis of drowning. It must be clearly explained, that intramuscular hemorrhages can occur under variant other circumstances linked to suffocation. Blunt external force must be excluded. The interpretation of such hemorrhages is difficult (see e.g. Püschel, K 1999).

4. The presence of fluid blood or blood with clots is, in cases of drowning, not appropriate to estimate the survival time, because fluid blood can occur as a result of drowning. (see Keil, W 2015 in Madea 2015 Rechtsmedizin). The length of the drowning process is thought to affect the expression of the drowning sights, detectable by autopsy (see Schneppe et al 2021). The passage must be changed or removed.

5. Please check in-depth that there is no correlation between temperature and age and after that please ad an suitable advice in your text.

6. Are there any correlations between the occurance of intramuscular hemorrhages and pathological findings or the medical history (e.g. cirrhosis of the liver or anticoagulants)?

7. Please check, that phrases in your text are in a logical order: i.e. "As a result, the frequency of intramuscular hemorrhage was significantly higher (p=0.0295) in cases aged ≥65 years than in those aged <65 years (Fig 3a, frequency, 0.24 vs. 0.41). The number of muscles with hemorrhage was also significantly higher (p=0.00945) in patients

128 ≥65 years than in those aged <65 years (Fig 3b, 0.38 vs. 1.0)." The order of the details the text 1. >65 years an 2. <65 years must be the same in the brackets.

8. The conclusion, that hemorrhages in the respiratory muscles of unidentified drowned bodies is an auxiliary finding of drowning and a useful finding for age estimation needs essentials revision. 1. These hemorrhages can not be understood as "real" drowning signs (see above). 2. Age estimation is a very challenging issue, that must be performed carefully and individually. Due to the fact, that these hemorrhages even occur your cases with younger victims, they are not helpful for age estimation in a particular case.

6. PLOS authors have the option to publish the peer review history of their article (what does this mean?). If published, this will include your full peer review and any attached files.

Reviewer #1: No

Reviewer #2: No

---

## [Author Response · Author response to Decision Letter 0]

2 Nov 2021

Dear Academic Editor and Reviewers,

We would like to thank you for your critical and instructive comments. We read the comments carefully and revised the manuscript in accordance with the reviewer’s comments, as follows. In the revised manuscripts, we indicate the modified points with highlights.

Answers to Editor

1. Following the editor's comment, we checked that our manuscript format conformed to the PLOS ONE’s format and corrected any non-conformities.

2. In accordance with the editor’s comment, we ensured that we have discussed whether all data/samples/autopsy reports were fully anonymized before we accessed them and/or whether the IRB or ethics committee waived the requirement for informed consent.

3. Following the editor's comment, I obtained an ORCID ID and clicked on the Fetch/Validate link next to the ORCID field in “Update my Information” page. 

4. Figure 1 shows a representative selection of autopsy photographs taken by the authors during the autopsy, with information on the sex and age of the study sample added to the legend of Figure 1 (please see page 6, lines136-138). The use of autopsy photographs in the article has been approved by the Ethics Committee as long as no individual is identified. Therefore, the images are not copyrighted.

Answers to Reviewer #1:

Major comments 

1. In accordance with your suggestion, we added discussion concerning the difference in intramuscular hemorrhage between freshwater and seawater drowning (please see page 13, lines 284-289 of Discussion).

2. We did not include cases from estuarine areas in our study sample because, as the reviewer stated, the mechanism of death in brackish water may be different from that in seawater or freshwater drowning. We are sorry for the omission in the original version of manuscript. We have added a note in the results section that cases in brackish water were excluded (please see page 8, lines 174-175 of Result).

Minor comments

1. In accordance with your suggestion, we set the month corresponding to every season (please see page 9, lines 194-196 of Result). 

2. In accordance with your suggestion, we changed “in patients” to “in victims” in new page 8, line 164. We have also changed the legend in Figure 3 to reflect this (please see legend to Fig 3, page 8, line 168).

Answers to Reviewer #2:

1. In accordance with your suggestion, we have included in the Introduction a more detailed description of the morphological findings suggestive of vital drowning and have added a description of the difficulties in diagnosing drowning death (please see pages 3-4, lines 46-76 of Introduction). We have also these contents to the Abstract, accordingly (please see page 2, line 23-25 of Abstract).

2. In reply to your suggestion, we noted that the usefulness of the plankton test is also debatable (please see page 4, lines 77-82 of Introduction).

3. Since we also believe that the possibility of blunt external force as a cause of the intramuscular hemorrhage assessed in this study should be ruled out, we have added descriptions in the Abstract (please see page 2, lines 38-41), Discussion (please see page 12, lines 263-266), and Conclusion sections (please see page 15, lines 333-336), respectively. We also added that intramuscular hemorrhage is also observed in other causes of death than drowning (please see page 12, lines 261-263).

4. In accordance with your suggestion, as it is inappropriate to estimate the elapsed time of drowning on the presence of fluid blood or blood with soft clots, we have deleted and modified those statements (please see page 14, lines 297-301). We have also deleted Suppl. Fig 2, accordingly.

5. In accordance with your suggestion, we examined the correlation between temperature and age and found no significant difference (please see page 5, lines 111-112 of Materials and methods, page 10, lines 214-215 of Results). Therefore, it is stated that low temperature and age are independent factors that predispose to intramuscular hemorrhage (please see page 2, lines 41-43 of Abstract, page 14, lines 310-311 of Discussion).

6. In accordance with your suggestion, we investigated the relationship between the occurrence of intramuscular hemorrhage and a potential bleeding tendency caused by disease (liver cirrhosis) and drugs (anticoagulants). The results showed that both the frequency of intramuscular hemorrhages and the number of muscles with hemorrhage were not affected by the presence or absence of bleeding tendency. This is a useful result and we have added a description of this to the Abstract (please see page 2, lines 36-38), Materials and methods (please see page 5, lines 100-101), Results (please see page 11, lines 231-236), new Fig 7 with legend (please see page 11, lines 238-240), and Discussion (please see page 15, lines 323-329, page 16, lines 340-341).

7. In reply to your suggestion, we modified the phrases in our text in a logical order (please see page 8, lines 163 and 165, page 10, lines 212 and 214, page 11, lines 225 and 226). 

8. In accordance with your suggestion, we have revised the first draft because the conclusions were overstated. We have kept these intramuscular hemorrhages to the extent that they give us a clue to differentiate between drowning death and other causes of death, if we can completely rule out the possibility of trauma or other causes of death (please see page 2, lines 38-41, page 15, lines 333-336). We deleted the mention of age estimation.

 We hope that the revised manuscript is found to be much improved and suitable for the publication in PLOS ONE. We thank you in advance for your great kindness and look forward to your favorable reply.

With my best regards,

---

## [Decision Letter · Decision Letter 1]

1 Dec 2021

Epidemiological analysis of intramuscular hemorrhage of respiratory and accessory respiratory muscles in fatal drowning cases

PONE-D-21-27906R1

Dear Dr. Hayashi,

We’re pleased to inform you that your manuscript has been judged scientifically suitable for publication and will be formally accepted for publication once it meets all outstanding technical requirements.

Kind regards,

Stephen Raverty

Academic Editor

PLOS ONE

Additional Editor Comments (optional):

Thank you for your attention and detailed responses to the reviewers. The text is considerably improved.

Reviewers' comments:

Reviewer's Responses to Questions

**Comments to the Author**

1. If the authors have adequately addressed your comments raised in a previous round of review and you feel that this manuscript is now acceptable for publication, you may indicate that here to bypass the “Comments to the Author” section, enter your conflict of interest statement in the “Confidential to Editor” section, and submit your "Accept" recommendation.

Reviewer #1: All comments have been addressed

Reviewer #2: All comments have been addressed

2. Is the manuscript technically sound, and do the data support the conclusions?

Reviewer #1: Yes

Reviewer #2: Yes

3. Has the statistical analysis been performed appropriately and rigorously? 

Reviewer #1: Yes

Reviewer #2: Yes

4. Have the authors made all data underlying the findings in their manuscript fully available?

Reviewer #1: Yes

Reviewer #2: Yes

5. Is the manuscript presented in an intelligible fashion and written in standard English?

Reviewer #1: Yes

Reviewer #2: Yes

6. Review Comments to the Author

Reviewer #1: Manuscript Number: PONE-D-21-27906

Title: Epidemiological analysis of intramuscular hemorrhage of respiratory and accessory respiratory muscles in fatal drowning cases

General comments

This manuscript has been well revised, both in text and figures, in accordance reviewers' comments. As a result, the content of this paper has been further organized and made easier to understand.

A reviewer consider this article has reached a level sufficient for adoption.

Reviewer #2: (No Response)

7. PLOS authors have the option to publish the peer review history of their article (what does this mean?). If published, this will include your full peer review and any attached files.

Reviewer #1: No

Reviewer #2: No

---

## [Editor Report · Acceptance letter]

6 Dec 2021

PONE-D-21-27906R1 

Epidemiological analysis of intramuscular hemorrhage of respiratory and accessory respiratory muscles in fatal drowning cases 

Dear Dr. Hayashi:

I'm pleased to inform you that your manuscript has been deemed suitable for publication in PLOS ONE. Congratulations! Your manuscript is now with our production department. 

Kind regards, 

on behalf of

Dr. Stephen Raverty 

Academic Editor

PLOS ONE